# Quality assessment of RNA in long-term storage: The All Our Families biorepository

Nikki L. Stephenson [1]*, Kylie K. Hornaday[2], Chelsea T. A. Doktorchik[1], Andrew W. Lyon[3], Suzanne C. Tough[1,4], Donna M. Slater[2,5]

1 Department of Community Health Sciences, Cumming School of Medicine, University of Calgary, Calgary, Alberta, Canada, 2 Department of Physiology and Pharmacology, Cumming School of Medicine, University of Calgary, Calgary, Alberta, Canada, 3 Department of Pathology and Laboratory Medicine, Saskatchewan Health Authority, Saskatoon, Saskatchewan, Canada, 4 Department of Paediatrics, Cumming School of Medicine, University of Calgary, Calgary, Alberta, Canada, 5 Department of Obstetrics and Gynecology, Cumming School of Medicine, University of Calgary, Calgary, Alberta, Canada

* nstephe@ucalgary.ca

## Abstract

### Background

The All Our Families (AOF) cohort study is a longitudinal population-based study which collected biological samples from 1948 pregnant women between May 2008 and December 2010. As the quality of samples can decline over time, the objective of the current study was to assess the association between storage time and RNA (ribonucleic acid) yield and purity, and confirm the quality of these samples after 7–10 years in long-term storage.

### Methods

Maternal whole blood samples were previously collected by trained phlebotomists and stored in four separate PAXgene Blood RNA Tubes (PreAnalytiX) between 2008 and 2011. RNA was isolated in 2011 and 2018 using PAXgene Blood RNA Kits (PreAnalytiX) as per the manufacturer's instruction. RNA purity (260/280), as well as RNA yield, were measured using a Nanodrop. The RNA integrity number (RIN) was also assessed from 5–25 and 111–130 months of storage using RNA 6000 Nano Kit and Agilent 2100 BioAnalyzer. Descriptive statistics, paired t-test, and response feature analysis using linear regression were used to assess the association between various predictor variables and quality of the RNA isolated.

### Results

Overall, RNA purity and yield of the samples did not decline over time. RNA purity of samples isolated in 2011 (2.08, 95% CI: 2.08–2.09) were statistically lower (p<0.000) than samples isolated in 2018 (2.101, 95% CI: 2.097, 2.104), and there was no statistical difference between the 2011 (13.08 μg /tube, 95% CI: 12.27–13.89) and 2018 (12.64 μg /tube, 95% CI: 11.83–13.46) RNA yield (p = 0.2964). For every month of storage, the change in RNA purity is -0.01(260/280), and the change in RNA yield between 2011 and 2018 is -0.90 μ g / tube. The mean RIN was 8.49 (95% CI:8.44–8.54), and it ranged from 7.2 to 9.5. The rate of

**Data Availability Statement:** The All Our Families questionnaire and medical record data are stored at Secondary Analysis for Generating Evidence (SAGE), a secure data repository managed by the

Alberta Centre for Child, Family and Community Research (https://policywise.com/resource/access-control-and-security/). Requests for this data (S01-197845.4) are welcomed. Data for this study are publicly available within the PRISM Dataverse: University of Calgary's Data Repository (doi: 10.5683/SP2/4JHR05).

**Funding:** This study, and the All Our Families Cohort are funded through Alberta Innovates (https://albertainnovates.ca) Interdisciplinary Team Grant (200700595, ST, DS, AL) and the Alberta Children's Hospital Foundation (http://www.childrenshospital.ab.ca/site/PageNavigator/about/about; RF-ABC001, ST). The funders had no role in study design, data collection and analysis, decision to publish, or preparation of the manuscript.

**Competing interests:** The authors have declared that no competing interests exist.

change in expected RIN per month of storage is 0.003 (95% CI 0.002–0.004), so while statistically significant, these results are not relevant.

## Conclusions

RNA quality does not decrease over time, and the methods used to collect and store samples, within a population-based study are robust to inherent operational factors which may degrade sample quality over time.

## Introduction

The All Our Families (AOF) study is a prospective pregnancy cohort from Calgary, Alberta. This study collected questionnaire, medical chart, and biological data (n = 1948 women), to better understand maternal and infant health, as well as research the biological, environmental and psychosocial determinants of adverse birth outcomes [1–3]. Maternal blood samples were collected using mobile phlebotomists between 2008 and 2011, and the success of recruitment for the study, as well as sample collection uptake, have been reported previously [3].

There are, however, several extraneous sources of bias which can influence the quality of the collected samples which are inherent to the operations of the study. Sample quality can be influenced by the time between sample collection and long-term storage, the ambient temperature at collection, efficiency of phlebotomists collecting samples, storage location (i.e. power fluctuations, freezer malfunctions) [4]. Additionally, the quality of biological samples can decline over time (e.g., RNA (ribonucleic acid) fragmentation) even if appropriate management and storage practices are maintained [5–7]).

The All Our Families cohort has collected biological samples from approximately 1900 pregnant women at two different time points during pregnancy (17–23 weeks and 28–32 weeks gestation), as well as cord blood at birth. These samples have been in long-term storage for 7 to 10 years, and the quality of the samples from the AOF in long-term storage have yet to be determined. From this study, we aimed to evaluate the RNA purity, yield, and integrity within these stored samples, and determine the potential factors which may influence the quality of the samples.

## Materials and methods

The AOF study is a prospective cohort study conducted in Calgary between August 2008 and July 2011 that aimed to assess maternal, perinatal and child outcomes [1, 2]. The cohort collected questionnaires assessing participant demographics, as well as psychosocial, clinical, obstetric, and behavioural data. Further, the cohort collected biological samples, including maternal whole blood, serum and plasma, and cord blood samples. To ensure consistency, and limit collection/storage variables, the cohort's biological samples were collected in a standardized method. A small pool of mobile phlebotomists collected the samples, placed them into tubes that stabilize intracellular RNA, and returned them to the lab for processing prior to long term storage in one of three freezers.

### Sample selection

Maternal whole blood samples (n = 1948 provided samples in total; n = 1862 provided samples at both time points during pregnancy) were stored in four separate PAXgene Blood RNA

Tubes (PreAnalytiX). In 2011, RNA was isolated, from two of the four PAXGene tubes (in each of n = 282 participants), for a collaborative study between the University of Calgary and the University of Toronto, to identify biomarkers for preterm birth by RNA array expression analysis [8]. In 2018, 1 of the remaining PAXGene tubes from each of the participants included in the 2011 study, were extracted for additional RNA expression studies (Fig 1). Forty-eight participants were excluded from the 2018 study due to missing data or withdrawal from the study. In 2020, 45 additional samples were isolated for an additional RNS expression study, and the RINs for these samples were also included in this study.

## RNA isolation and measurement

RNA was isolated from maternal whole blood samples with the PAXgene Blood RNA Kit [9] (PreAnalytiX, Qiagen/BD) in both 2011 and 2018. Optical absorbency ratios (260/280), a measure of RNA purity, and RNA yield were measured using a NanoDrop [10] (Thermo Fisher Scientific). The technicians performing sample extractions in both 2011 and 2018 received training and supervision through the same (Slater) lab, following the identical sample isolation protocol in order to reduce technical variance. On another sample (group 1, n = 164 from 5 months-25; group 2 n = 45 from 111–130 months storage time; n = 209), an RNA integrity number was determined via RNA 6000 Nano Kit and Agilent 2100 BioAnalyzer; Agilent Technologies, Santa Clara, CA [11].

## Statistical analysis

The purity and yield of the extracted RNA were compared using paired t-tests, and response feature linear regression. The association of RIN with storage time was assessed using linear regression using STATA 15 IC statistical software. Variables included in the response feature analysis and regression models are described in Table 1. Each potential predictor was assessed as a potential modifier and confounder through backwards elimination model fitting using

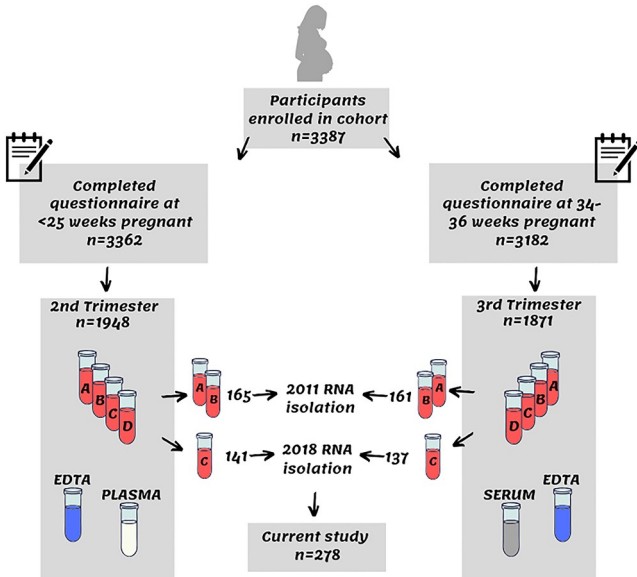

**Fig 1. Sample collection flow diagram.** Flow diagram of biological sample collection, and study sample inclusion.

**Table 1. Variable summary of potential predictors of RNA quality.**

| Variable | | Extraction Year | Mean | 95% CI |
|---|---|---|---|---|
| Storage time (months) | Group 1 | 2011 | 12.34 | 8.60–16.09 |
| | | 2018 | 99.64 | 95.12–104.16 |
| | Group 2 | 2020 | 121.71 | 120.33–123.10 |
| **Variable** | | **Minimum** | **Maximum** | **Median** |
| Temperature at collection (˚C) | | -12.5 | 16.2 | 3.1 |
| Time in -20˚C storage (days) | | 0 | 55 | 2 |
| **Variable** | | | **Frequency (n)** | **Percent (n/278)** |
| Phlebotomist | | A | 21 | 10.05% |
| | | B | 50 | 23.92% |
| | | C | 56 | 26.79% |
| | | D | 68 | 32.54% |
| | | E | 7 | 3.35% |
| | | F | 7 | 3.35% |
| Collection | | Second trimester | 141 | 50.72% |
| | | Third trimester | 137 | 49.28% |
| Freezer location | | A | 157 | 56.47% |
| | | B | 110 | 39.57% |
| | | C | 11 | 3.96% |

CI, confidence interval; n, sample size

likelihood ratio tests and assessing the influence of predictors on the primary outcomes, RNA yield, purity and RIN.

Assumptions of the distributional form for linear regression were assessed visually through graphing a "q-q plot" (residuals vs standard normal quantiles), and assessment of constant variance was assessed visually through graphing the residuals vs fitted values. Further, the assumptions of statistical independence within the purity and yield comparisons are addressed through analyzing the change in outcomes rather than the outcomes by year.

## Ethical considerations

Data used for identifying participants and participant samples were stored on the 256-bit encrypted server at the University of Calgary. The All Our Families study was approved by the Child Health Research Office, Alberta Health Services, and the Conjoint Health Research Ethics Board of the University of Calgary. Written informed consent was obtained from the study participants at the time of recruitment, who were also provided copies for their records. All modifications to incorporate the current study were reported and approved (REB 15–0248). All procedures were conducted in accordance with ethical principles and the Helsinki Declaration of 1975 (2008 revision) [12]. Data analysis was conducted using de-identified data, and therefore, all necessary privacy precautions were implemented.

## Results

The 260/280 ratios from the RNA isolated in 2011 ranged from 1.88 to 2.28, with a mean ratio of 2.07 (95% CI: 2.06–2.07). The 260/280 ratios from the RNA isolated in 2018 ranged from 1.98 to 2.21, with a mean ratio of 2.10 (95% CI: 2.07–2.13). None of the samples were found to be of low quality or show evidence of contamination, as determined by 260/280 ratios [13].

**Table 2. Summary of RNA quality outcomes.**

| Outcome | Extraction Year | Mean | 95% CI |
|---|---|---|---|
| Optical density of RNA (260/280) | 2011 | 2.08 | 2.07–2.09 |
| | 2018 | 2.10 | 2.10–2.10 |
| RNA yield (ug per tube) | 2011 | 13.08 | 12.27–13.89 |
| | 2018 | 12.64 | 11.83–13.46 |
| RIN | n/a | 8.44 | 8.38–8.49 |

CI, confidence interval

The yield of RNA isolated in 2011 ranged from 1.32 to 39.16 μg with a mean ratio of 13.08 (95% CI:12.27–13.89) per PAXgene tube. The RNA isolated in 2018 ranged from 0.068 to 39.51, with a mean ratio of 12.64 (95% CI: 11.86–13.46) per PAXgene tube. The detailed results pertaining to RNA yield and purity outcomes are outlined in Table 2, and all samples were within the acceptable optical absorbency ratio range (above 1.8 [10]) for use in downstream analysis.

The paired t-test comparing means for the purity of RNA in the 2011 and 2018 RNA isolations, does provide evidence that there is a difference in mean RNA purity by extraction year (p<0.000), with an increase in mean RNA purity within the 2018 extraction year (Fig 2). The paired t-test comparing means for the yield of RNA in the 2011 and 2018 RNA isolations, indicates there was no difference in RNA yield by extraction year (p = 0.2964) (Fig 3).

Response feature analysis using linear regression provides evidence towards an association between storage time and the purity of RNA. Effect measure modification due to phlebotomist, storage locations, time in -20˚C storage before transportation to -80˚C, trimester of collection, and outdoor temperature on the day of collection were all assessed during model fitting, using

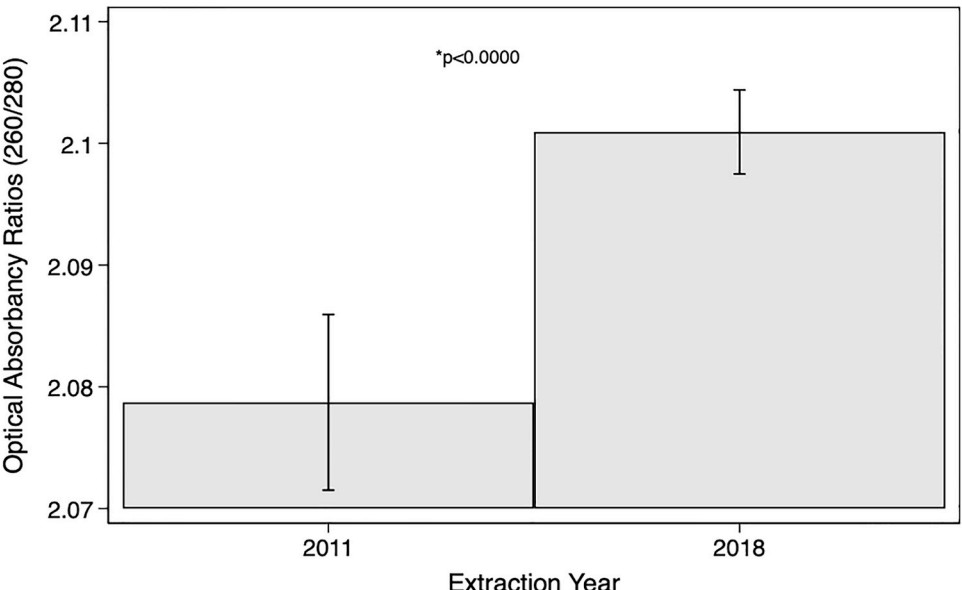

**Fig 2. RNA purity t-test results.** Paired t-test comparing mean RNA purity (260/280 optical absorbency ratios) in the 2011 and 2018 RNA isolations.

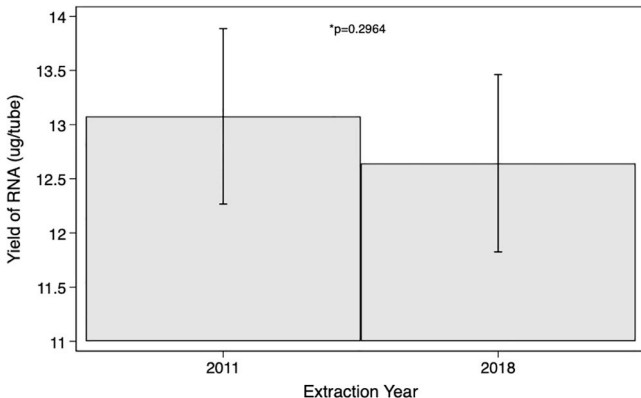

**Fig 3. RNA yield t-test results.** Paired t-test comparing mean RNA yield in the 2011 and 2018 RNA isolations.

a likelihood ratio test which compared maximum likelihood estimates of the nested models. None of these factors modified the association between the change in isolated RNA purity and length of storage (p = 0.9330). All of the above factors were subsequently assessed as potential confounders using backwards elimination. None of these factors led to a meaningful difference in the estimate of the association between storage time and purity of RNA; therefore the crude linear regression model most accurately describes the relationship between storage time and RNA purity. The presented model shows that the rate of change of the mean RNA 260/280 ratio (2018-260/280 minus 2011-260/280) is -0.0098076 (95% CI: -0.0120756, -0.0075395) per month of storage, and as the 95% confidence interval does not enclose the null value we conclude that there is a statistically significant association between storage time and RNA purity (Fig 4), though it is unlikely that this positive association is relevant as all measures remain within the acceptable range for purity.

Evaluation of the mean difference in the yield of RNA between 2011 and 2018 through response feature analysis using linear regression also provides sufficient evidence that there is an association between storage time and yield of RNA. Effect measure modification of phlebotomist, storage locations, time in -20˚C storage before transportation to -80˚C, trimester of collection, and outdoor temperature on the day of collection were all assessed during model fitting, using a likelihood ratio test which compared maximum likelihood estimates of the nested models. None of these factors modified the association between the change in isolated RNA yield and length of storage (p = 0.3337). All of the above factors were subsequently assessed as potential confounders using backwards elimination. None of these factors led to a meaningful difference in the estimate of the association between storage time and yield of RNA; therefore the crude linear regression model most accurately describes the relationship between storage time and yield of RNA. The presented model shows that the rate of change of the mean difference in yield of RNA (2018 minus 2011) is -0.90 μg /tube (95% CI:-1.14, -0.67) per month of storage, and as the 95% confidence interval does not enclose the null value we conclude that there is a statistically significant association between storage time and change in yield of RNA between extraction times, however, this would not translate to a relevant difference (Fig 5).

Using linear regression the relationship between RIN and storage time, over 5–130 months, provides sufficient evidence that the RIN does not decline over time. Effect measure modification of phlebotomist, storage locations, time in -20˚C storage before transportation to -80˚C, trimester of collection, and outdoor temperature on the day of collection were all assessed

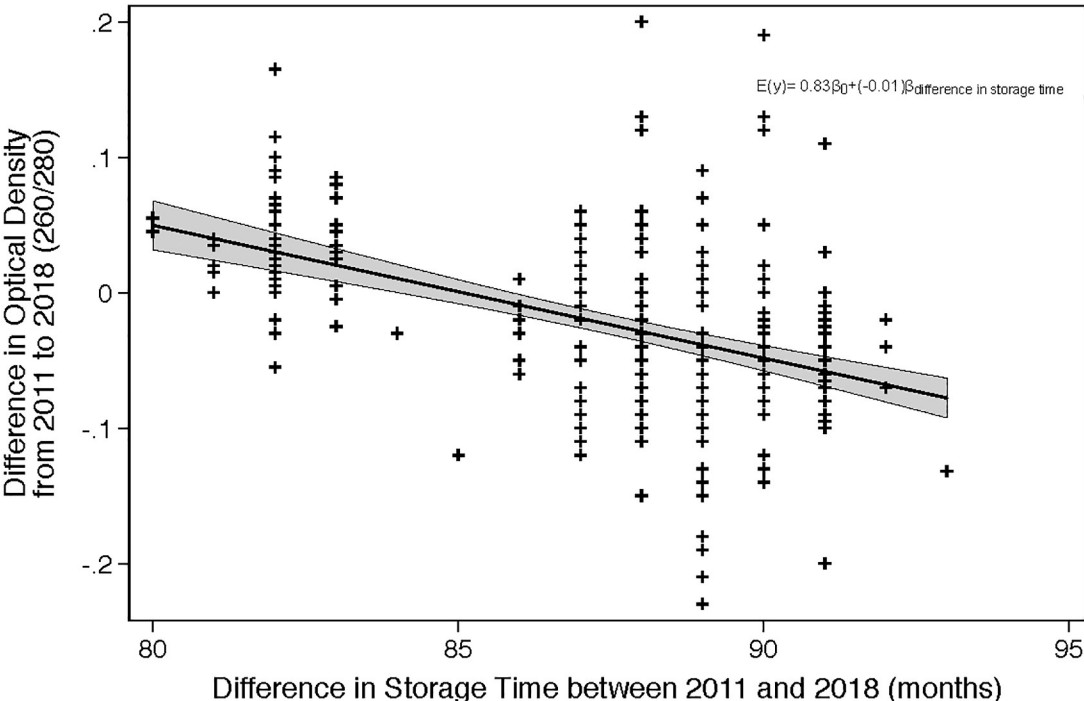

$E(y)= 0.83\beta_0+(-0.01)\beta_{\text{difference in storage time}}$

**Fig 4. RNA purity linear regression results.** Association between the difference in RNA purity (2011 260/280 ratio minus 2018 260/280 ratio) and length of storage time. The negative slope implies that the difference between 2011 and 2018 increased with storage time, with the 2018 purity increasing over time.

during model fitting, using a likelihood ratio test which compared maximum likelihood estimates of the nested models. None of these factors modified the association between the change in isolated RNA yield and length of storage (p>0.05). All of the above factors were subsequently assessed as potential confounders using backwards elimination. Of these factors only phlebotomist and outdoor temperature on the day of collection lead to a meaningful difference in the estimate of the association between storage time and RIN; therefore the linear regression model was adjusted for these confounding factors. The mean RIN was 8.49 (95% CI:8.44–8.54), and it ranged from 7.2 to 9.5. RIN results from the early group (5–25 months) and the later group (111–130 months) were modelled separately, and then combined to ensure the groups did not introduce heterogeneity. The presented model using all available RIN data shows that the rate of change in expected RIN per month of storage is 0.003 (95% CI 0.002–0.004), so again while statistically significant, these results are not relevant (Fig 6).

## Discussion

The current study evaluated the quality of RNA from a prospective cohort study in Calgary, Alberta, comparing RNA isolated from maternal whole blood samples isolated in 2011 with matched samples isolated in 2018. This study demonstrated that the yield and purity of RNA isolated from the maternal whole blood samples remained high throughout long-term storage regardless of extraneous factors which are inherent to cohort operations. The paired student's t-test showed that there was no difference in RNA yield between 2011 and 2018 and an minimal increase in RNA purity over time; however regression analysis suggests that this increase in both RNA yield, purity, and integrity overtime is not relevant as all values remain within the acceptable ranges for use.

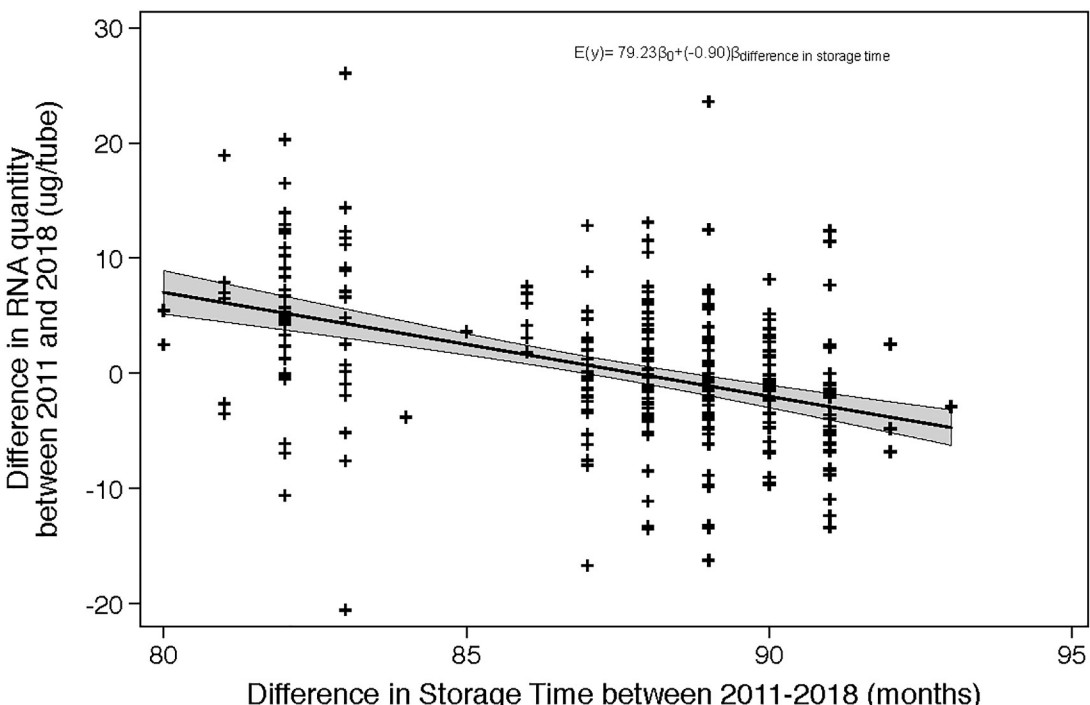

**Fig 5. RNA yield linear regression results.** Association between the difference in RNA quantity (2011 μg /tube minus 2018 μg /tube) and length of storage time. The negative slope implies that the difference between 2011 and 2018 increased with storage time, with the 2018 quantity increasing over time.

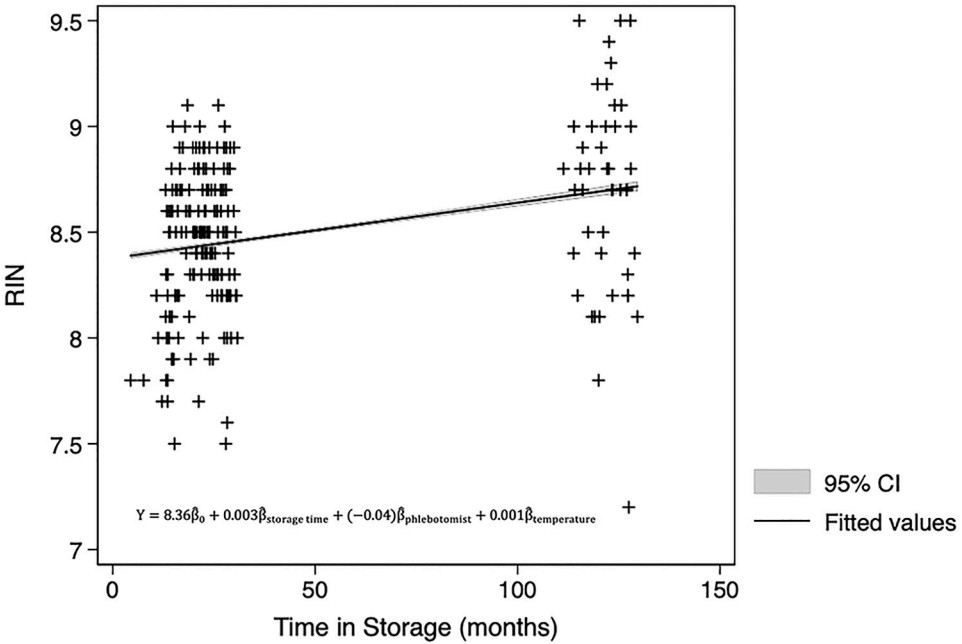

**Fig 6. RNA integrity linear regression results.** Association between the RNA integrity number and length of storage time.

The 260/280 absorbance ratios of the RNAs all fell within the acceptable range; therefore, the RNA was deemed to be pure. Although the purity was shown to be statistically different between samples isolated in 2011 versus 2018, it would seem that the purity of RNA increased over time. The narrow confidence intervals at the two time points indicate the overall precision of the estimates. The observed statistical difference in the mean optical absorbency ratio as assessed through the paired t-test may be attributable to technician precision when performing the RNA isolation. The paired t-test comparing means for the yield of RNA in the 2011 and 2018 RNA isolations, failed to provide evidence that there was a difference in RNA yield by extraction year. The width of the calculated confidence intervals resulting from the t-test implies a lack of precision; potentially due to the RNA isolation, the measurement of RNA yield, or differences inherent to the sample or participant. However, neither of these comparisons take extraneous factors, such as time in storage, into consideration.

Therefore, phlebotomist who collected the sample, sample storage location, time in -20˚C storage before transportation to -80˚C, trimester of sample collection, and outdoor temperature on day of collection were considered as potential predictors for the association between the change in RNA purity (2011 260/280–2018 260/280) and difference in storage time between 2011 and 2018. These same factors were considered when assessing the between the change in RNA yield (2011 μg /tube– 2018 μg /tube) and the difference in storage time between 2011 and 2018.

For every month of storage, the change in RNA purity (where the difference is calculated as 2011 ratios minus 2018 ratios) is -0.01 (260/280), which implies that the RNA purity is improving with storage time. For every month of storage, the change in RNA yield between 2011 and 2018 is -0.90 μg, which implies that the RNA yield is also increasing with storage time. As with the paired t-test analysis for RNA purity, the observed association in the expected optical absorbency ratio and yield may be attributable to increased precision when performing the RNA isolation (indicated by the visual assessment of Figs 4 and 5), or potentially residual confounding which was not addressed in the current analysis.

RNA integrity was further assessed to confirm the quality of RNA using a subset of samples over a different time period. While it would seem that there is a miniscule increase in RIN over time, the visual assessment of the regression model indicates that there is an increase in precision of the estimates rather than an actual increase in RNA integrity. As with the sample's purity, all RIN estimates were above 7, which is within the acceptable range for most downstream applications [8].

The authors' recognize that it is biologically implausible that the quantity, purity, and integrity of RNA would increase over time, however it is also recognized that precision of techniques likely increases over time. Collinearity between storage time and the order of assessment is inherent to the study design, in that the first sample to be assessed for purity, yield, and RIN will likely have the shortest duration. We conclude that, though it appears to be an increase in quality over time, this is due to the increased precision via increased skill of the technician. However, we can still conclude that there is no reduction in purity, yield or integrity over storage time, as one can see that if the precision (an unmeasured variable) were to remain constant throughout all estimates for purity, yield, and integrity would remain within acceptable clinical ranges.

In addition to the maternal whole blood samples (four PAXGene tubes per time point), the AOF study also collected maternal serum (n = 1858) at 17–23 weeks gestation and plasma (n = 1947) at 17–23 and 28–32 weeks gestation, and cord blood (n = 1439) were obtained at delivery. The AOF study further collected whole blood samples stored in EDTA tubes for DNA isolation (n = 1944 at time point 1 and n = 1857 at time point 2). To date only the maternal whole blood within the current study and that of Heng et al. have been utilized for RNA

isolation; therefore the authors' conclusions are isolated to these RNA samples, which in most cases there remain four PAXgene tubes stored and available for study.

The range of samples collected by the AOF cohort enables the extraction and research of various proteins, RNA, and DNA that can be used for future research studies investigating pregnancy and maternal and child health. As an additional strength, the AOF study further collected questionnaire data that assessed various obstetrical and clinical information, psychosocial parameters (e.g., mental health assessments and social support), and demographic characteristics. These comprehensive questionnaires offer the opportunity to assess physiology during pregnancy and childbirth, in conjunction with various parameters such as mental and clinical health.

This methods paper described the quality of the cohort's whole blood samples concerning the RNA quality and confirmed that the quality of biological samples was not significantly influenced by those factors intrinsic to cohort operations but may contribute to RNA deterioration. The current study assures the integrity of studies which have previously used these samples [8], as well as informs future investigators of the quality of the cohort's biological materials. This research also emphasizes the importance of proper storage and maintenance of biological samples, as well as informs best practices in maternal whole blood collection and storage for large population-based studies.

## Acknowledgments

The authors acknowledge the contribution and support of AOF team members, the Slater lab, and our participants. We are grateful to the investigators, coordinators, research assistants, graduate and undergraduate students, volunteers, clerical staff, lab technicians and managers. We are extremely grateful to all the women who provided samples for this study.

## Author Contributions

**Conceptualization:** Nikki L. Stephenson, Donna M. Slater.

**Data curation:** Nikki L. Stephenson, Kylie K. Hornaday, Chelsea T. A. Doktorchik.

**Formal analysis:** Nikki L. Stephenson.

**Funding acquisition:** Suzanne C. Tough, Donna M. Slater.

**Investigation:** Kylie K. Hornaday, Chelsea T. A. Doktorchik.

**Methodology:** Andrew W. Lyon, Suzanne C. Tough.

**Project administration:** Nikki L. Stephenson.

**Resources:** Andrew W. Lyon, Suzanne C. Tough, Donna M. Slater.

**Supervision:** Suzanne C. Tough, Donna M. Slater.

**Writing – original draft:** Nikki L. Stephenson.

**Writing – review & editing:** Kylie K. Hornaday, Chelsea T. A. Doktorchik, Andrew W. Lyon, Suzanne C. Tough, Donna M. Slater.

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
