## [Decision Letter · Decision Letter 0]

20 Sep 2019

PONE-D-19-21994

Quality assessment of RNA in long-term storage: The All Our Families biorepository

PLOS ONE

Dear Ms Stephenson,

Thank you for submitting your manuscript to PLOS ONE. After careful consideration, we feel that it has merit but does not fully meet PLOS ONE’s publication criteria as it currently stands. Therefore, we invite you to submit a revised version of the manuscript that addresses the points raised during the review process.

We would appreciate receiving your revised manuscript by Nov 04 2019 11:59PM. To enhance the reproducibility of your results, we recommend that if applicable you deposit your laboratory protocols in protocols.io, where a protocol can be assigned its own identifier (DOI) such that it can be cited independently in the future. For instructions see: http://journals.plos.org/plosone/s/submission-guidelines#loc-laboratory-protocols

We look forward to receiving your revised manuscript.

Kind regards,

Rania Mohamed Labib, PhD

Academic Editor

PLOS ONE

Journal Requirements:

Reviewers' comments:

Reviewer's Responses to Questions

**Comments to the Author**

1. Is the manuscript technically sound, and do the data support the conclusions?

Reviewer #1: Partly

Reviewer #2: Partly

2. Has the statistical analysis been performed appropriately and rigorously? 

Reviewer #1: I Don't Know

Reviewer #2: No

3. Have the authors made all data underlying the findings in their manuscript fully available?

Reviewer #1: Yes

Reviewer #2: Yes

4. Is the manuscript presented in an intelligible fashion and written in standard English?

Reviewer #1: Yes

Reviewer #2: Yes

5. Review Comments to the Author

Reviewer #1: The authors Claim (in the title and conclusion) to have studied quantity and quality of RNA isolated from blood stored in PAXgene Blood RNA Tube for 7-10 years. However, they present data on RNA quantity and puritiy (Ratio 260/280 nm).

Purity of RNA is no measure of quality, which should be depicted as integrity. A more or less degraded RNA - mark of pure quality - may sow perfect purity. RNA quality should therefore be assessed e.g. by BioAnalyser or - if not available - electrophoresis. To make it even better, expression levels of a set of genes or Array Analysis will give even better evidence of good oerall and mRNA quality.

The authors demonstrate statistically significant differences in purity of RNA isolated in 2011 and 2018, with purity rising over time, which is an overvaluation of statistics. This difference is probably statistically significant, but it is not relevant. A pure nucleic acid is generally considered to have an 260/280 ratio of around 2, which is the case at both time points.

The conclusions the authors draw from their results are correct: RNA quantity and purity do not change with storage time of blood samples under the given conditions. But before concluding that the remaining samples show unchanged good quality for further studies this should be demonstrated by methods mentioned above.

Reviewer #2: Nikki Stephenson et. al, in their manuscript ‘Quality assessment of RNA in long-term storage: The All Our Families biorepository’, addressed a very general but important issue of sample storage in biorepository. In this manuscript the authors have analyzed RNA from whole blood samples from a cohort from 2011 and compared its quality with the RNA samples from the cohort of 2018. The authors conclude that RNA quality does not decrease over long term storage and the conditions used for the collection of samples, storage and extraction are robust. Although the manuscript hold merit with the high sample number from the biorepository, I have following specific concerns about the manuscript.

Specific comments

1. The RNA concentration and purity are measured on simple nano-drop; which are not specific and usually give variable results. It is necessary to have RNA integrity number (RIN) by a bioanalyzer which. Further, correlation of RIN to RNA Quality Score (RQS) can further enhance the authenticity of the data.

2. The conclusions drawn from the statistical analysis are biased, supporting the fact that RNA quality and purity are either better or improving with time. Although the authors themselves have mentioned that it is clinically insignificant.

6. PLOS authors have the option to publish the peer review history of their article (what does this mean?). If published, this will include your full peer review and any attached files.

Reviewer #1: No

Reviewer #2: No

---

## [Author Response · Author response to Decision Letter 0]

4 Nov 2019

1. Editor’s comment: “Please ensure that your manuscript meets PLOS ONE's style requirements, including those for file naming.”

Authors’ response: Thank you for the comments. To the best of the authors’ knowledge this submitted revised manuscript adheres to the PLOS ONE's style requirements. The style of in-text citation has been altered to include square brackets and are utilizing the Vancouver style for referencing. Also, the files have been renamed as suggested.

2. Reviewer #1 comment: “The authors Claim (in the title and conclusion) to have studied quantity and quality of RNA isolated from blood stored in PAXgene Blood RNA Tube for 7-10 years. However, they present data on RNA quantity and puritiy (Ratio 260/280 nm).”

Authors’ response: Thank you for your comments. We have adjusted each of the methods (line 115), and discussion (lines 244-260) surrounding RNA concentration and RNA optical absorbency to reflect when we are specifically discussing each. Also, we have relabelled each of the figures and tables to note specifically when we are discussing RNA concentration (yield) and RNA optical absorbency (purity). 

3. Reviewer #1 comment: “Purity of RNA is no measure of quality, which should be depicted as integrity. A more or less degraded RNA - mark of pure quality - may sow perfect purity. RNA quality should therefore be assessed e.g. by BioAnalyser or - if not available - electrophoresis. To make it even better, expression levels of a set of genes or Array Analysis will give even better evidence of good oerall and mRNA quality.”

Authors’ response: We understand that a 260/280 ratio is a measure of purity, however our consideration of quality was to discuss both RNA concentration and purity together. To address your comment we have included an analysis on a subset of samples for which we have RNA integrity numbers. Though this is a smaller sample size (n=164) over a shorter time span (25 months), we hope that this inclusion (lines 48-51, 73, 102-104, 107-108, 111, Table 2, 193-205, Figure 6, and 251-267), along with the previous response feature analysis of RNA concentration and purity will provide sufficient evidence for our conclusions that neither the quality nor quantity of RNA decreased over storage time.

4. Reviewer #1 comment: “The authors demonstrate statistically significant differences in purity of RNA isolated in 2011 and 2018, with purity rising over time, which is an overvaluation of statistics. This difference is probably statistically significant, but it is not relevant. A pure nucleic acid is generally considered to have an 260/280 ratio of around 2, which is the case at both time points.”

Authors’ response: We agree that the observed increase in both RNA quantity and quality, though statistically significant, are not clinically important. However, we do not feel that this is an overvaluation of statistics, rather than this one piece of statistical evidence provided by the regression model output is driven primarily by an increase in precision of the estimates, and when the data is assessed visually through figures 4-6 this increase in precision can be seen. We have mentioned both in the results section (lines 165-166, 186-187, 204-205), as well as in the discussion section (lines 217, 244-260) that while the analyses show a statistically significant positive association between storage time and both purity and concentration, this association is not clinically significant. However we can conclude that there was no decrease in purity, concentration, or RIN over time.

5. Reviewer #1 comment: “The conclusions the authors draw from their results are correct: RNA quantity and purity do not change with storage time of blood samples under the given conditions. But before concluding that the remaining samples show unchanged good quality for further studies this should be demonstrated by methods mentioned above.”

Authors’ response: At the request of the reviewer we have included further analysis on a subset of samples analysing the association between storage time and RIN (lines 48-51, 73, 102-104, 107-108, 111, Table 2, 193-205, Figure 6, and 251-267). However, we are unable at this time to assess the RIN after the same amount of storage as we were assessing with purity and yield due to logistic constraints.

6. Reviewer #2 comment: “The RNA concentration and purity are measured on simple nano-drop; which are not specific and usually give variable results. It is necessary to have RNA integrity number (RIN) by a bioanalyzer which. Further, correlation of RIN to RNA Quality Score (RQS) can further enhance the authenticity of the data.”

Authors’ response: Thank you for your comments. We have included further analysis on a subset of samples analysing the association between storage time and RIN. Though this is a smaller sample size (n=164) over a shorter time span (25 months), we hope that this inclusion (lines 48-51, 73, 102-104, 107-108, 111, Table 2, 193-205, Figure 6, and 251-267) along with the previous response feature analysis of RNA concentration and purity will provide sufficient evidence for our conclusions that the quality of RNA does not decrease over storage time. Unfortunately, the authors’ did not have access to the electropherogram trace for each of these samples, thus the height of the 18s and 28s RNA peaks, the area under the 18s and 28s RNA peaks, and the fast region area are unknown. Without these values one cannot compute the RNA quality score (Copois et al, 2007 https://doi.org/10.1016/j.jbiotec.2006.07.032). As the reviewer noted in the comment, RIN and RQS have been correlated, and the algorithm proposed by Schroeder et al, 2006 (doi:10.1186/1471-2199-7-3) has long been used as a gold standard of RNA quality, therefore although the authors’ recognize the value of including the RQS we feel that the inclusion of the RIN does enhance the authenticity of the data even without the RQS.

7. Reviewer #2 comment: ‘The conclusions drawn from the statistical analysis are biased, supporting the fact that RNA quality and purity are either better or improving with time. Although the authors themselves have mentioned that it is clinically insignificant.’

Authors’ response: While the authors’ agree that the systematic error leading to increased precision over time would bias the resultant estimates produced via both the response feature analysis and the linear regression, we feel that our conclusions- that the purity, yield, and RIN, do not decrease as a function of storage time- are unbiased as we clearly state in our description of the results that the resultant positive association is not clinically significant (lines 165-166, 186-187, 204-205), and in our discussion (lines 217, 244-260) we further emphasize this point noting that it is attributable to increased precision as seen through visual assessment of the models in figures 4 through 6, and potentially residual confounding. To accentuate this we have included an additional paragraph within the discussion (lines 244-260) outlining the influence of precision on the analysis and we offer our interpretation of our analysis.

---

## [Decision Letter · Decision Letter 1]

11 Mar 2020

PONE-D-19-21994R1

Quality assessment of RNA in long-term storage: The All Our Families biorepository

PLOS ONE

Dear Ms Stephenson,

Thank you for submitting your manuscript to PLOS ONE. After careful consideration, we feel that it has merit but does not fully meet PLOS ONE’s publication criteria as it currently stands. Therefore, we invite you to submit a revised version of the manuscript that addresses the points raised during the review process.

Please address the additional comments from reviewer 1.

We would appreciate receiving your revised manuscript by Apr 25 2020 11:59PM. To enhance the reproducibility of your results, we recommend that if applicable you deposit your laboratory protocols in protocols.io, where a protocol can be assigned its own identifier (DOI) such that it can be cited independently in the future. For instructions see: http://journals.plos.org/plosone/s/submission-guidelines#loc-laboratory-protocols

We look forward to receiving your revised manuscript.

Kind regards,

Zhong-Cheng Luo

Academic Editor

PLOS ONE

Reviewers' comments:

Reviewer's Responses to Questions

**Comments to the Author**

1. If the authors have adequately addressed your comments raised in a previous round of review and you feel that this manuscript is now acceptable for publication, you may indicate that here to bypass the “Comments to the Author” section, enter your conflict of interest statement in the “Confidential to Editor” section, and submit your "Accept" recommendation.

Reviewer #1: (No Response)

2. Is the manuscript technically sound, and do the data support the conclusions?

Reviewer #1: Yes

3. Has the statistical analysis been performed appropriately and rigorously? 

Reviewer #1: I Don't Know

4. Have the authors made all data underlying the findings in their manuscript fully available?

Reviewer #1: Yes

5. Is the manuscript presented in an intelligible fashion and written in standard English?

Reviewer #1: Yes

6. Review Comments to the Author

Reviewer #1: The authors now corrected RNA quality for puritiy and quantity, and give plausible explanations for the apparent increase in purity over time. These changes are statistically significant, but not clinically relevant. "Clinically significant" should be changed for "clinically relevant", or even better for "relevant", since this is rather a technical than a clinical aspect. Passages discussion the statistically evident seeming improvements should be shortened.

Furthermore, to address RNA qualitiy, the authors included RIN numbers for a subset of samples (please change "subsample" for "subset of samples" , line 244), demonstrating no drop in overall RNA quality. Unfortunately, the storage time of the samples tested for RIN ranges only over 24 month as compared to 94 months for the purity and quantity measures. It would be nice to see RIN results at least for a small subset of samples after medium and long term storage (eg. 50 and 90 months.

7. PLOS authors have the option to publish the peer review history of their article (what does this mean?). If published, this will include your full peer review and any attached files.

Reviewer #1: No

---

## [Author Response · Author response to Decision Letter 1]

8 Sep 2020

Reviewers' comments:

The authors now corrected RNA quality for puritiy and quantity, and give plausible explanations for the apparent increase in purity over time. These changes are statistically significant, but not clinically relevant. "Clinically significant" should be changed for "clinically relevant", or even better for "relevant", since this is rather a technical than a clinical aspect. Passages discussion the statistically evident seeming improvements should be shortened.

Furthermore, to address RNA qualitiy, the authors included RIN numbers for a subset of samples (please change "subsample" for "subset of samples" , line 244), demonstrating no drop in overall RNA quality. Unfortunately, the storage time of the samples tested for RIN ranges only over 24 month as compared to 94 months for the purity and quantity measures. It would be nice to see RIN results at least for a small subset of samples after medium and long term storage (eg. 50 and 90 months.

Authors’ response:

The Authors’ thank Reviewer 1 for the additional comments. 

As per request we have altered our terminology “clinically significant” to “technically relevant”, and we have decreased the discussion concerning statistical significance. Please see lines 50, 182, 204, 228, 240, and 281.

To address the second part of the comment, we have included an additional 45 RINs with a long-term storage of 111-130 months. We are unable to complete an analysis at 50-90 months, as all samples had been stored past that time. 

We modeled the short-term storage (11-25 month) RINs and the long-term storage (111-130 months) RINs separately to ensure we would not introduce heterogeneity into a combined model. The most parsimonious model including all RIN data is reported, with the same conclusion as previously reported. We hope that this addition increases the reviewers confidence in the results regarding sample integrity.

---

## [Editor Report · Decision Letter 2]

22 Sep 2020

PONE-D-19-21994R2

Quality assessment of RNA in long-term storage: The All Our Families biorepository

PLOS ONE

Dear Dr. Stephenson,

Thank you for submitting your manuscript to PLOS ONE. After careful consideration, we feel that it has merit but does not fully meet PLOS ONE’s publication criteria as it currently stands. Therefore, we invite you to submit a revised version of the manuscript that addresses the points raised during the review process.

We look forward to receiving your revised manuscript.

Kind regards,

Zhong-Cheng Luo

Academic Editor

PLOS ONE

Additional Editor Comments (if provided):

There are 2 tracking-changes version in the same downlowned R2 PDF file, confusing !

Line 103, delete "a" before"another'.

Table 2: please present all values to the precision of 2 decimal points. 3 decimal points are unnecessary.

Line 50, Line 166 and Line 208, delete "technically". The word "relevant" is sufficient, and please do use the expression "technicically relevant" throughout the paper.

Line 167, delete the confusing "(greater than for purity"

---

## [Author Response · Author response to Decision Letter 2]

26 Oct 2020

Comment: There are 2 tracking-changes version in the same downlowned R2 PDF file, confusing !

Response: This has been corrected.

Comment: Line 103, delete "a" before"another'.

Response: The text has been removed.

Comment: Table 2: please present all values to the precision of 2 decimal points. 3 decimal points are unnecessary.

Response: The table values have been changed to accommodate this request.

Comment: Line 50, Line 166 and Line 208, delete "technically". The word "relevant" is sufficient, and please do use the expression "technicically relevant" throughout the paper.

Response: The term has been changed throughout the document.

Comment: Line 167, delete the confusing "(greater than for purity"

Response: The text has been removed.

---

## [Editor Report · Decision Letter 3]

29 Oct 2020

PONE-D-19-21994R3

Quality assessment of RNA in long-term storage: The All Our Families biorepository

PLOS ONE

Dear Dr. Stephenson,

Thank you for submitting your manuscript to PLOS ONE. After careful consideration, we feel that it has merit but does not fully meet PLOS ONE’s publication criteria as it currently stands. Therefore, we invite you to submit a revised version of the manuscript that addresses the points raised during the review process.

We look forward to receiving your revised manuscript.

Kind regards,

Zhong-Cheng Luo

Academic Editor

PLOS ONE

Additional Editor Comments (if provided):

Page 9, line 185, the donfidence interval does not make sense : "-0.090 μg /tube (95% CI:-1.14, -0.67)", please check and correct the numbers.

Edits:

Page 9, line 194, replace "deline" with "decline"

Page 10, line 206, replace "0.03" with "0.003"
---

## [Author Response · Author response to Decision Letter 3]

29 Oct 2020

Editors comment: Page 9, line 185, the donfidence interval does not make sense : "-0.090 μg /tube (95% CI:-1.14, -0.67)", please check and correct the numbers.

Authors’ Response: This has been corrected to -0.90 μg /tube (95% CI:-1.14, -0.67)

Editors comment: Page 9, line 194, replace "deline" with "decline"

Authors’ Response: This has been corrected.

Editors comment: Page 10, line 206, replace "0.03" with "0.003"

Authors’ Response: This has been corrected.

---

## [Editor Report · Decision Letter 4]

3 Nov 2020

Quality assessment of RNA in long-term storage: The All Our Families biorepository

PONE-D-19-21994R4

Dear Dr. Stephenson,

We’re pleased to inform you that your manuscript has been judged scientifically suitable for publication and will be formally accepted for publication once it meets all outstanding technical requirements.

Kind regards,

Zhong-Cheng Luo

Academic Editor

PLOS ONE
---

## [Editor Report · Acceptance letter]

20 Nov 2020

PONE-D-19-21994R4 

Quality assessment of RNA in long-term storage: The All Our Families biorepository 

Dear Dr. Stephenson:

I'm pleased to inform you that your manuscript has been deemed suitable for publication in PLOS ONE. Congratulations! Your manuscript is now with our production department. 

Kind regards, 

on behalf of

Dr. Zhong-Cheng Luo 

Academic Editor

PLOS ONE